# Factors shaping Covid-19 vaccine acceptability among young people in South Africa and Nigeria: An exploratory qualitative study

**Marisa Casale** [1,2]*, **Oluwaseyi Somefun** [1], **Genevieve Haupt Ronnie** [3], **Joshua Sumankuuro** [1,4,5], **Olagoke Akintola** [1], **Lorraine Sherr** [6], **Lucie Cluver** [2,7]

**1** School of Public Health, University of the Western Cape, Western Cape, South Africa, **2** Department of Social Policy and Intervention, University of Oxford, Barnett House, Oxford, United Kingdom, **3** Centre for Social Science Research, University of Cape Town, Cape Town, South Africa, **4** Department of Public Policy and Management, Faculty of Public Policy and Governance, Simon Diedong Dombo University of Business and Integrated Development Studies, Wa, Ghana, **5** School of Nursing, Paramedicine and Healthcare Sciences, Faculty of Science and Health, Charles Stuart University, Bathurst, New South Wales, Australia, **6** University College London, London, England, United Kingdom, **7** Department of Psychiatry and Mental Health, University of Cape Town, Cape Town, South Africa

* maj.casale@icloud.com

## Abstract

Covid-19 vaccine hesitancy among young people can be seen as an acute – but not isolated – phenomenon within an alarming longer-term trend of broader vaccine distrust in Africa. Yet there are still considerable knowledge gaps in relation to the scope and drivers of low vaccine acceptability among young people. Moreover, better frameworks and tools are needed to conceptualise and better understand acceptability in this population group. We applied the recently published *Accelerate Framework for Young People's Acceptability* to guide qualitative research with young people living in South Africa and Nigeria. We aimed to investigate their overall acceptability of the Covid-19 vaccine, and explore factors shaping this acceptability and willingness to be vaccinated. In collaboration with seven community-based organisation partners, we conducted 12 in-person focus groups and 36 remote interviews with 163 individuals aged 15-24. Through a collaborative, iterative process we conducted thematic analysis, incorporating aspects of both deductive and inductive approaches. Our findings show how vaccine acceptability is shaped by a multiplicity of inter-related factors. They also provide a more in-depth perspective of some of these phenomena, their relative importance and their connections in this group of young people. Limited vaccine understanding, conflicting information and distrust, the influence of others, and fear of side effects were key inter-related drivers of low vaccine acceptability. Factors promoting Covid-19 vaccine acceptability were instead: positive perceptions of vaccine safety and efficacy, protection from disease, protection of others, and a desire to return to normal activity. We discuss implications of these findings for policy and practice, both to increase acceptability of Covid-19 vaccination among young people, and more broadly promote vaccination as a critical component of public health programs. Lastly, we reflect on this first application of the *Accelerate* Framework, and implications for its use in future studies.

**Data availability statement:** The authors are unable to make the full dataset publicly available due to institutional ethical requirements and the need to protect the confidentiality of partner organizations. However, relevant excerpts from the qualitative data transcripts may be provided upon reasonable request, subject to the signing of a confidentiality agreement. Requests for data access can be directed to Prof. Brian Van Wyk, an institutional representative from the School of Public Health at the University of the Western Cape, at bvanwyk@uwc.ac.za, who will oversee external requests. The theoretical model guiding this study has been previously published in an open access format.

**Funding:** This work was supported by the UKRI GCRF Accelerating Achievement for Africa's Adolescents (Accelerate) Hub (ES/S008101/1 to LC and LS); the European Research Council (ERC) under the European Union's Horizon 2020 research and innovation programme (771468 to LC); UNICEF-ESARO (to LC); and the Oak Foundation (OFIL-20-057 to LC). The funders had no role in study design, data collection and analysis, decision to publish, or preparation of the manuscript.

**Competing interests:** The authors have declared that no competing interests exist.

## Introduction

Low acceptability and uptake of vaccines is a key threat to the success of public health responses to infectious disease outbreaks, in Africa and globally. These issues have recently gained particular attention in relation to the Covid-19 response [1,2]. The unprecedented nature of this crisis, as well as high levels of vaccine hesitancy [3,4], have brought to the fore the importance of acceptance and uptake - as well as availability - of biomedical tools for public health responses.

The concept of vaccine acceptability is closely related to vaccine hesitancy, with the former referring to positive cognitive and behavioural responses to the vaccine [5,6] and the latter referring to a delay, reluctance or refusal in uptake [7,8]. As articulated by various authors, these concepts should be considered distinct from constructs related to intervention engagement, such as willingness or intention to engage with an intervention, intervention uptake and retention. Constructs related to intervention engagement may, however, predict or be predicted by acceptability [5,9]. Both low vaccine acceptability and vaccine hesitancy have in fact been associated with lower uptake of vaccines and other interventions [5,10].

Low vaccine acceptance remains a challenge and ongoing concern for the Covid-19 response, as we continue to live with the disease and the uncertainty of future variants [2,4]. However, it is a challenge that does not start or end with Covid-19. Vaccine hesitancy has been reported as a major obstacle for public health safety in Africa prior to the pandemic in relation to other conditions [11–14] and is likely to remain an obstacle for future national and international responses to infectious disease crises. In particular, the past decade has marked a global trend of growing vaccine mistrust and resistance, including in Africa [15] and specifically South Africa [16–18]. Despite vaccine availability, many individuals have delayed or refused vaccines for themselves or their children, even when vaccines were available, thus exposing communities to various infectious diseases [19].

Covid-19 vaccine hesitancy and broader vaccine hesitancy could also be interlinked and each potentially have a negative effect on the other [17]. There is some evidence to suggest that the experience of Covid-19 may have further fuelled vaccine hesitancy, as opposed to encouraging vaccination [11,20]. For example, the South African Social Attitudes Survey (SASAS) found that beliefs in serious side effects of vaccines had increased since national Covid-19 lockdowns, while the Africa CDC found that about 20% of participants were less inclined to vaccinate than before the pandemic [17]. A review of South African surveys concluded that "Covid-19 vaccine hesitancy may be the tip of the iceberg of general vaccine hesitancy in South Africa." [17] p. 929.

Fortunately, vaccine acceptance has been shown to be variable and therefore a potentially modifiable factor that is responsive to interventions [17,21]. Cooper et al [17], for example, argue that variations in levels of acceptance of Covid-19 vaccination over time in South Africa may reflect the volatility of perceptions in a context of rapidly changing knowledge about the disease and strategies to manage it. Similar dynamics have been documented in global surveys [2]. However, effectively intervening to increase vaccine acceptability and uptake requires not only understanding the extent of the phenomenon but also what shapes it: specifically, "the often complex and multi-layered issues" (Cooper 2021b, p. 930) or factors that encourage and hinder acceptability of vaccines.

Research to date, among general populations in Africa, has suggested that there may be some common factors driving low vaccine acceptability across countries and diseases. These include low levels of education and awareness, inefficient government efforts, fear of side effects, poor routine-vaccine history, belief in conspiracy theories and misinformation on social media [1,22,23]. Yet, there are still considerable information gaps in relation to the scope, determinants and causal mechanisms of vaccine hesitancy in Africa [11,13] and,

in particular, among young people. This is salient since young people under the age of 25 account for almost 60% of Africa's population, highlighting the huge potential of this population group to thrive and contribute positively to society, despite health and social challenges [24]. Multiple surveys suggest lower levels of vaccine acceptance among adolescent and youth populations, both in South Africa and in higher income countries [17,25–27]. Young people may also be more at risk for COVID-19 infection and transmission due to their frequent engagement in social activities, both within and outside of school, and lower likelihood of consistently adhering to preventative measures [28,29].

Research with adolescents and youth in sub-Saharan Africa has highlighted possible determinants for Covid-19 vaccine hesitancy such as inadequate information and understanding, conflicting information, religious beliefs, a low perceived risk of contracting Covid-19, concerns about vaccine safety and effectiveness, and low trust in public health institutions [30–32]. However, most of the available in-depth studies with young people utilize small sample sizes within specific demographic cohorts [8,33,34]. Interventions have used health education, incentives, legal requirements for vaccination and education-based delivery with various efficacy and little adjustment to local contexts [14]. Additionally, understanding acceptability and vaccine uptake among young people presents unique challenges. Unlike younger children or adults, decisions regarding vaccination may be more intricate and less straightforward. These decisions could be 'mediated,' where parents or guardians play a significant role in the decision-making process, adding layers of complexity to understanding and addressing vaccine hesitancy among young people [35].

Given the importance of vaccine acceptability among young people, and limitations of the literature and interventions to date, this area would benefit from better conceptual models and further mixed-methods data collection, with young people themselves, to drive understanding, intervention and broad approaches to future vaccine initiatives [36,37]. Existing theoretical models, including the WHO proposed '5C' vaccine hesitancy model [38] and Sekhon et al's Theoretical Framework of Acceptability [5], are based primarily on research from high income countries. Some authors have argued that these tools omit key construct dimensions and prioritize individual over social processes [11,13,39]. Moreover, our extensive search of the literature did not yield any tools for framing or assessing vaccine acceptability that had been validated in Africa specifically with young people [13,40].

In this paper, we apply a framework we recently developed for intervention acceptability among young people, to investigate acceptability of the Covid-19 vaccine 'intervention' among young people in Africa. 163 individuals aged 15-24, living in South Africa and Nigeria, participated in this research, which is part of a broader exploratory qualitative study. The objectives of the broader study were to apply our *Accelerate Framework for Young People's Acceptability* with young people living in Africa, in order to investigate their acceptability of both the Covid-19 vaccine (as an 'intervention' that could be considered common and relevant to all young people across Africa) and the various health and skills building interventions they were receiving from local community-based organisations (CBOs). The purpose of this analysis was to determine young people's overall acceptability of the Covid-19 vaccine, and to explore the factors influencing this acceptability and their willingness (or unwillingness) to be vaccinated.

## Methods

### Theoretical framework

Our study was guided by a recently published conceptual framework we developed [6], the *Accelerate Framework for Young People's Acceptability* [6]. To develop this framework we drew from Sekhon et al's [5] Theoretical Framework for Acceptability (TFA) and further review

and formative work, conducted over a period of 2 years within an international adolescent research Hub [39,40]. Our model development process is described in detail elsewhere [6]. Our definition of 'young people' for the model development process and this paper refers to individuals between 10 and 24 years of age, comprising the partly overlapping categories of adolescents (10-19) and youth (15-24).

The *Accelerate Framework for Young People's Acceptability* proposes nine components of acceptability, defined as "a multi-faceted construct that reflects the extent to which people delivering or receiving a healthcare intervention consider it to be appropriate, based on anticipated or experienced cognitive and emotional responses to the intervention" [5]. These are: affective attitude, intervention understanding, perceived positive effects, relevance, perceived social acceptability, burden, ethicality, perceived negative effects and self-efficacy. Each of these components are illustrated and defined in Fig 1 [6]. The *Accelerate Framework* was used to conceptualize the study, develop the data collection instrument and guide the data analysis.

## Context

**Covid-19 vaccination among young people in South Africa and Nigeria.** During the COVID-19 pandemic, South Africa implemented a comprehensive vaccination campaign that evolved over time, starting with healthcare workers and vulnerable populations in early 2021[41], and expanding to the general population, including adolescents aged 12-17, by

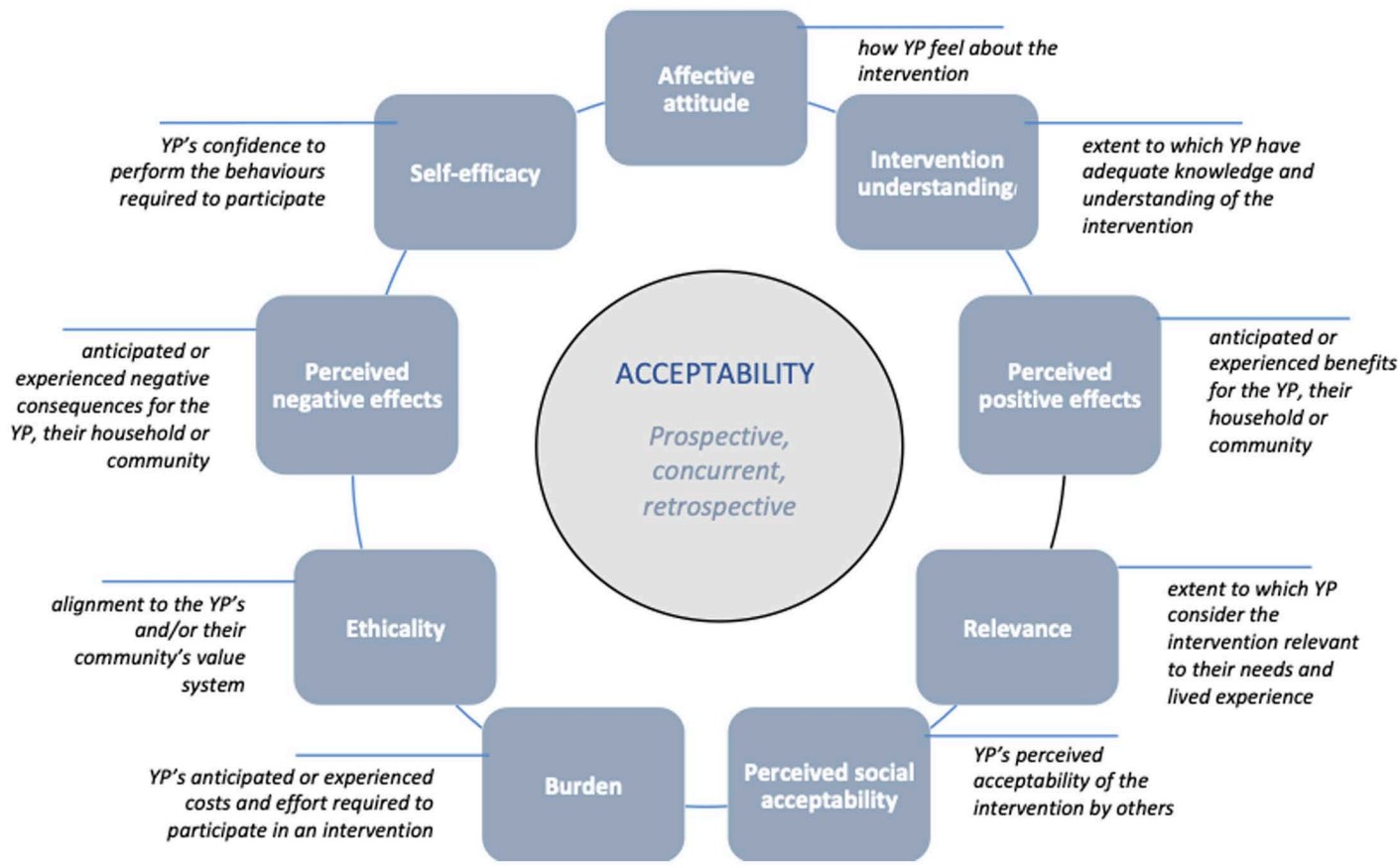

**Fig 1. Accelerate Framework for Young People's Acceptability.**

early 2022 [42]. As of October 2022, in South Africa, only 38% of adolescent girls and 30% of adolescent boys aged 12–17 had received at least one dose of a COVID-19 vaccine [43].

The country established an extensive network of vaccination sites across urban and rural areas to ensure broad access, supported by robust public communication efforts to address vaccine hesitancy and engage younger individuals. This campaign's reach and infrastructure was commended but challenges such as vaccine hesitancy and distribution disparities were noted [44]. Comprehensive vaccination records allowed for effective tracking and evaluation of the campaign's impact [45].

Nigeria's COVID-19 vaccination program started in March 2021, and by September 21, 2022, only 15% of the eligible population were fully vaccinated, due to issues like poor cold-chain management, government distrust, and communication failures [46]. During the COVID-19 pandemic, young people in Nigeria exhibited a mix of apprehension, resilience, and adaptability in response to the crisis [47]. The initial stages of the pandemic were marked by widespread uncertainty and fear, particularly due to misinformation and limited access to reliable information about the virus. Many young people, especially those in urban areas, quickly adapted to the new norms of social distancing, mask-wearing, and remote learning, though these measures were challenging to implement in more rural or densely populated communities [48]. When COVID-19 vaccines became available, the reception among young people was varied. While some were eager to get vaccinated as a means to protect themselves and return to a semblance of normalcy, others were hesitant due to prevalent misinformation, skepticism about the vaccine's safety, and mistrust in government and health institutions [48]. Public health campaigns, peer education, and the involvement of trusted community figures played crucial roles in improving vaccine acceptance [49].

**CBO partners.** We worked with a total of seven partner community-based organisations (CBOs) providing sexual and reproductive health (SRH) or skills development interventions among young people within our age group of interest: three in South Africa and four in Nigeria. In Nigeria, one of the CBOs offered training programs in Kano State; these were aimed at improving the livelihoods of under-served young people through digital inclusion and life skills training. A second CBO partner ran a youth-focused advocacy project in Benue State, aimed at providing a safe space to discuss HIV and SRH issues. Additionally, some participants were from a pilot project at a university research institute in Ibadan; this project trained adolescents on COVID-19 prevention measures and encouraged them to develop personal projects to help combat the pandemic. We also interviewed young adults participating in a government SRH program in Edo State; this program aimed at empowering adolescents and youth to achieve optimal health and development goals, and to make contraceptives more accessible.

In South Africa, our Cape Town-based CBO partner offered a 12-month digital skills training program to young people, as well as paid internships in the industry. The remaining two CBO partners were based in South Africa's KwaZulu-Natal province. One had been working in rural communities in the greater Durban area for over 70 years to promote health and wellbeing, and, at the time of the research, was implementing both an early child development program to empower caregivers (including adolescent and young adult caregivers), and a youth training centre offering courses in digital and other life skills. The second CBO was a longstanding national organisation working in urban and semi-urban areas in KwaZulu-Natal; it provided youth-focused HIV prevention and SRH programmes in group settings.

## Sampling

Since this was an exploratory study, conducted with limited resources and within a limited timeframe, a convenience sampling approach was employed. The researchers contacted CBOs

working with young people within the desired age group in South Africa and Nigeria, with whom they already had contacts and/or a working relationship. Those organisations who responded positively, and were able to collaborate with the research team over the timeframe available for the field research, partnered with the researchers for this study. The CBOs worked directly with the researchers to recruit young people from their respective programs.

We conducted 12 in-person focus groups and 36 remote interviews with a total of 163 young people aged 15-24 living in South Africa (n=136) and Nigeria (n= 27), between 10th January and 31 March 2022. In KwaZulu-Natal, South Africa, our two CBO partners' staff worked with the researchers to directly recruit young people for focus groups. For the remaining focus groups and all IDIs, the CBO representatives made contact with the young people to ascertain their overall willingness to participate and be contacted by the researchers, after which the researchers contacted them directly. The researchers went through lists of potential participants provided by the CBOs, and contacted these one by one until they had reached the desired numbers of young people who were willing to participate in the research. The aim was to recruit as equal as possible a number of male and female participants within the indicated age range.

## Data collection

The choice of methods used to collect data (in-person FGDs versus remote IDIs) was decided with the CBO partners working in each area, based on factors such as the location of the researchers, access and logistics factors, and the preferences and availability of the participants themselves. For example, at the time of the research, all authors were based in South Africa, which presented logistical challenges for conducting group discussions in Nigeria. Additionally, our convenience sampling approach, which relied on referrals through existing CBO networks, made it challenging to gather young people into online focus groups. Given the above and considering the diversity of the participants' backgrounds, including those from Northern (East and West) and South-Western Nigeria, we opted for in-depth interviews. Similarly, for our Cape Town-based sample a decision was made to conduct remote individual interviews, as opposed to focus groups; this decision was based on the preferences of the CBO partner and the participants themselves, as well as challenges encountered in finding a venue and time that would suit most participants. Some of the benefits of IDIs were, however, confidentiality of information shared and the ability to potentially capture the nuanced and varied perspectives of participants.

Thirty-six IDIs were conducted with young people in South Africa (9), and Nigeria (27). These in-depth interviews lasted for approximately 60 minutes and were conducted by two of the authors in English, via virtual platforms such as WhatsApp call and Zoom, based on participants' preferences. The Nigerian participants were asked whether they would prefer to conduct the interviews in their local language as opposed to English, however all participants expressed comfort with interviews in English. Individuals who participated remotely in the in-depth interviews received compensation for airtime.

Instead in KwaZulu-Natal, focus groups appeared to be a feasible and suitable option, given the location and availability of the lead researcher to reach the various research sites, the geographical location of the participants, the more limited access to and experience with multimedia (particularly in deep rural areas), and the logistical support that the CBO partners could make available. The focus groups were conducted in person, with 127 young people from urban and rural locations, in the local language (isiZulu). The CBO partners facilitated participant recruitment, the provision of venues for the sessions and (in the case of deeper rural communities) transport for the researchers to these venues. The FGDs were held in proximity to the communities from which young people were recruited. In some cases, where

possible, they were held on the CBO partners' premises; in deeper rural communities, instead, they were held at community spaces or homesteads, in agreement with community members. These sessions ranged from one to two hours in duration and were interrupted with a snack break to reduce participant fatigue. Group participant numbers ranged from 8 to 15. The focus group discussions were conducted by two bilingual field researchers, with the support and oversight of the lead author. The field researchers had extensive previous training and experience in conducting research with young people and in research ethics; they were also trained by the lead author over a two-week period specifically for this project. Participants were provided with a hot meal after the sessions and reimbursed for their transport expenses.

Topic guides were used to guide focus group discussions, and participants were given score cards to rate the vaccine 'intervention'. IDIs were guided by semi-structured question-naires following the same format as the topic guides. Based on a review of the conceptual and empirical acceptability literature we used two approaches to explore acceptability through our semi-structured data collection instruments: 1) an initial question asking young people to 'rate' the Covid-19 vaccine 'intervention' on a scale of 1 to 5 (1 being the lowest and 5 being the highest rating). This was followed by a subsequent open question asking participants to justify their rating; and 2) more specific questions based on the nine respective components included in our *Accelerate* framework. For example, for the 'intervention understanding' component we asked the following: *Can you tell me whether you have a good understanding of the Covid-19 vaccine, what it is and how it works? Do you feel you have been provided with sufficient information about the vaccine? What do you think can be done to improve under-standing for you and other young people like yourself?* Similar questions were asked for each framework component.

For accuracy and analysis, the in-depth interviews were recorded, transcribed, and then quality checked by two of the authors. The focus group discussions were transcribed, trans-lated into English, and subsequently quality checked by the bilingual field researchers.

Ethics approval to conduct this research was received by the University of the Western Cape's Biomedical Research Ethics Committee (BM21/10/39). Formal (written) permission was also provided by the executive boards of the two KwaZulu-Natal CBO partners. Written consent was obtained from all participants, and in the case of participants younger than 18, also from their parents or guardians. Consent forms were translated into and explained in the local languages for participants that could not speak English.

## Data analysis

We analysed the data through a collaborative group process. The analysis was conducted with the support of Dedoose software, which allowed team members to work simultaneously online and from different locations. The main analytical approach employed was template analysis, a form of thematic analysis [50–52]. The data were analyzed by means of an iterative process incorporating aspects of both deductive and inductive approaches, as described below [53–55].

The data analysis was conducted by four of the authors, in consultation with the bilingual interviewers who conducted the focus groups in KwaZulu-Natal. Transcripts were divided among the four researchers on the team. All researchers read all the transcripts, without coding, to familiarize themselves with the data and make note of what stood out as interesting. Each researcher then coded three transcripts, of which at least one from a focus group. Initial codes were identified from the transcript content, in terms close to the language used by respondents, and attention was paid to the way the codes were treated and presented [56].

An initial broad coding template was first developed deductively, based on the nine component constructs of the acceptability theoretical framework, that had also informed the design of the topic guides and interview schedules. These nine constructs were used as initial base (overarching construct) themes in the coding structure. Potential subthemes within these broader base themes were also identified. The team then met virtually to compare and discuss their initial list of generated codes, and cluster these within the nine base themes. Codes that did not seem to fit well within this coding template were 'parked' for further discussion. After a refined coding template was agreed on, this was uploaded as a code tree using Dedoose software, with agreed definitions for each code to guide the coders. The researchers then divided the remaining transcripts to complete the coding process. Regular virtual meetings were held over a period of two months, to identify, refine and review the base and sub-themes, resolve challenges and iteratively refine the analytical template.

While we paid attention to areas of disagreement in the data, the study design, sub-sample sizes and breadth of the analysis, did not allow us to properly explore differences in perceptions across race, gender, countries or (urban versus rural) locations, as explained further in the limitations section below.

### Reflexivity

Throughout the research process, we employed several reflexivity strategies to critically examine and manage our potential biases. Each researcher maintained a reflexive journal, where we regularly documented our thoughts, assumptions, and emotional responses during data collection and analysis. These journals helped us to continuously reflect on how our backgrounds and perspectives might influence our interactions with participants and the interpretation of data. Additionally, we held regular reflexive discussions in team meetings, where we collectively explored our positionalities and their potential impact on the research process. For the KwaZulu-Natal-based research, these discussions involved the bilingual field researchers who had conducted the focus groups; their feedback and impressions were recorded at the end of each day in the field following debriefings with the lead researcher. These strategies ensured that we remained aware of our subjectivities, allowing us to present findings that are more grounded in the participants' experiences.

## Results

### Participant characteristics

S1 Table shows the demographic characteristics of participants. Although we aimed for a 50/50 male to female participant ratio, the majority of participants were female, since young women were more like to be participating in the selected CBO programs and to be willing to participate in the research. For the FGDs for example, only 26 of the 127 participants were male and the remaining 101 were female. Eight FGDs with 86 young people were conducted in a deep rural area in KwaZulu-Natal, while the remaining four FGDs were conducted with 41 young people residing in urban areas. Approximately half of the FGD participants (64) were between 15 and 19 years of age, while the remaining participants (63) were 20-24 years old. Sixty-four of the 127 focus group participants (around 50%) were studying, Thirty-five (28%) reported having children, and only 3 (around 2%) said they were employed.

For the in-depth interviews (IDIs), just over half of the participants (56%) were female. Most (89%) participants were between 20-24 years and had all completed secondary education. About 47% were unemployed, 19% self-employed and 25% employed. All IDI participants, both in Nigeria and South Africa (Cape Town), resided in urban areas. None of the IDI participants reported having children. It is clear from this demographic information that IDI participants were overall older than the FGD participants, better educated and had more work opportunities, in part as a result of their (more urban) geographical location.

## Themes and subthemes

All emerging themes could be incorporated within the nine components of our acceptability framework, so no additional overarching construct or base themes were included. However, multiple subthemes were identified within these base themes, as illustrated in Fig 2 and discussed in more detail below.

1. Affective attitudes

Affective attitudes refer broadly to how young people feel about the intervention, including their overall and/or 'gut' feel towards it. The large majority of adolescent and youth participants demonstrated a negative attitude towards the Covid-19 vaccine. This was reflected in the scores given when young people were asked to rate the vaccine 'intervention' on a scale of 1 to 5 (with 1 being the lowest and 5 the highest score): 40 young people chose '1', 20 gave a score of '2', 43 gave a score of '3'. Only 24 of the 127 FGD participants gave the vaccine a score of 4 or 5 (15 and 9 respectively). The main reasons emerging to explain these negative attitudes were distrust and fear, although some young people also spoke about their unhappiness at being 'forced' to vaccinate. Less dominant views, among a minority of young people, were instead represented by positive attitudes towards the vaccine, and indifference or uncertainty.

**Negative attitude towards the vaccine: Distrust, fear and unhappiness about 'forced' vaccination.** Distrust was the dominant theme explaining negative attitudes towards the vaccine, and was often related to conspiracy theories. This derived from various factors,

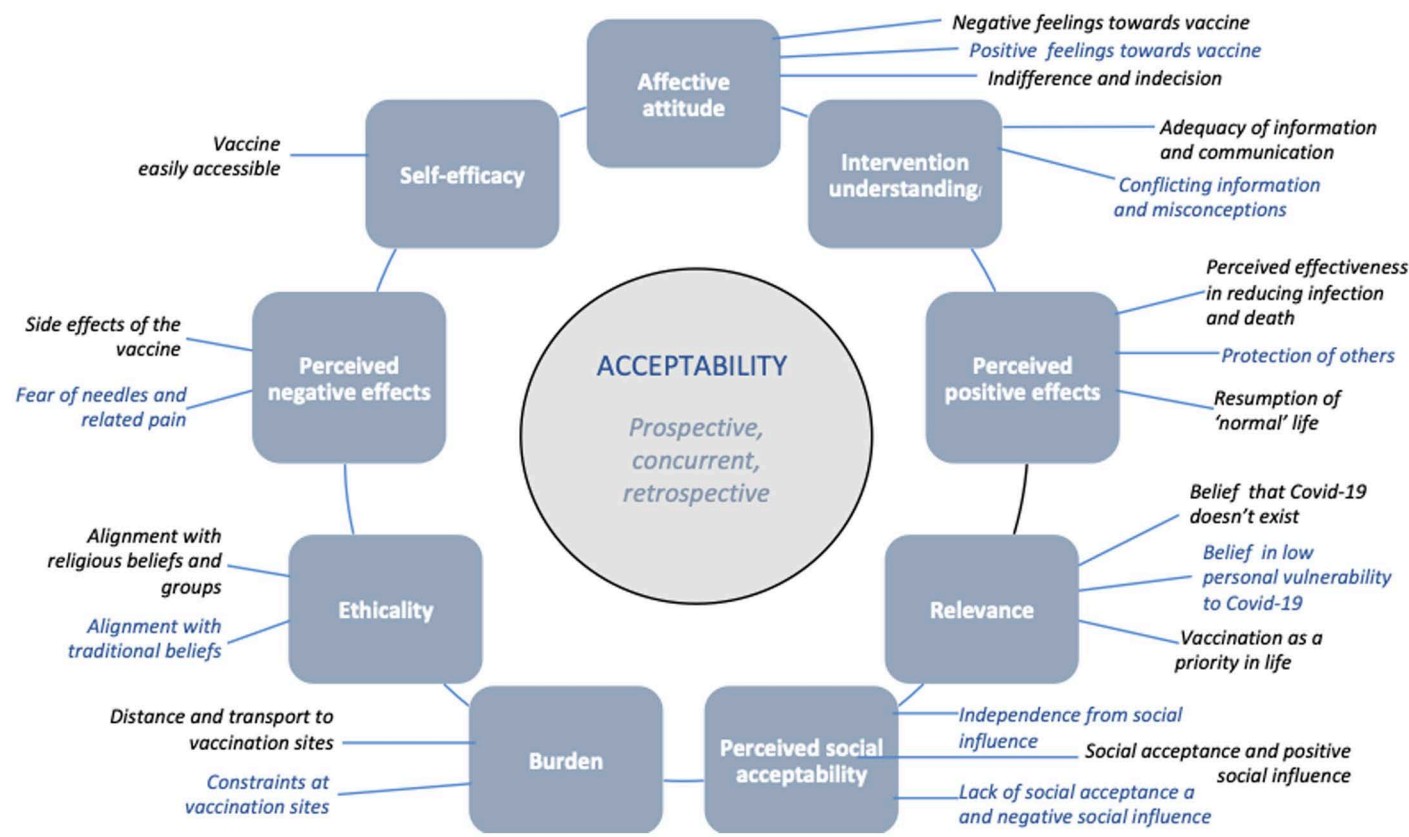

**Fig 2. Final thematic framework depicting base themes and subthemes.**

including: the fact that vaccines were seen as 'freebies' received from other countries; distrust of government, politicians and scientists; the pace at which the vaccine had been produced; and observed or believed side effects.

> *I once had a video footage where it depicts the vaccine dosage having a smallest size tracker. The tracker is meant to be injected into the body of those who are getting vaccinated. They say the tracker is there to monitor your movements by the government*
>
> *(Male FGD participant, Urban, South Africa).*

Distrust was also, at times, linked to specific information sources, such as social media, and to conflicting information and misunderstandings of how the vaccine worked.

> *The thing is, it is the people who scare us a lot. On social media platforms, you see a lot there, people are posting negative things which are scary.*
>
> *(Female FGD participant, Rural, South Africa).*

The second most frequently raised negative response to the vaccine was fear, which was in some cases closely linked to distrust. In many cases this fear resulted from observed or believed side effects, such as becoming ill and even dying because of the vaccine, and in some cases from conspiracy theories.

> *I feel frightened because there are people who get sick after taking the vaccine. I am thinking what if I decide to take the vaccine then I get side effects or die, this frightens me.*
>
> *(Female FGD participant, Rural, South Africa).*

Some young people were simply afraid of the procedure, for example of receiving an injection.

> *The problem with me is I am just scared of anything called injection, any sharp instrument.*
>
> *(Female FGD participant, Rural, South Africa).*

A number of young people spoke about their objections to what they considered 'obligatory' or 'forced' vaccination, since it had become a pre-requisite to access venues, including workplaces and learning institutions. For some, there was also a sense of betrayal, given the messaging that being vaccinated was voluntary.

> *I feel, once they say, if you don't do your vaccine, you won't do, we won't allow you to go into so and so place, you won't do this. You are creating more fear. Instead encourage people and tell them "okay, if you take your vaccine and you want to work here and you want to do your IT here, you are protecting yourself and others"... Because once you make it like a requirement, people would start finding a way around it, just to get to that place.*
>
> *(Female IDI participant, Urban, Nigeria).*

**Positive attitude towards the vaccine.** A minority of young people expressed positive attitudes towards the Covid-19 vaccination. This was motivated mainly by a recognition of its effectiveness in reducing Covid-19 infection and related death, and the importance of feeling protected and protecting (particularly elderly) family members and others. A few young people referred to the opportunities it allowed for in terms of access to places

and opportunities. In some cases, these positive attitudes were also a result of not having experienced many negative side effects or having observed worrying side effects among others who had been vaccinated.

> *Now that I knew, I feel safe and yes, I know that even though the virus comes in contact with me now, I won't be like, I won't be able to get down or have any symptoms or any effects since, I've been vaccinated, so I feel safe... there's a particular criteria to some places, like if you apply for something online... I feel ok, I feel excited about that I'm vaccinated.*
>
> *(Male IDI participant, Urban, Nigeria)*

**Indifference and indecision.** A third group (minority) of young people had neither strong negative or positive attitudes towards the Covid-19 vaccine, but instead felt indifferent; reasons included not fully understanding the vaccine, believing that it was (at times or always) ineffective.

> *I don't feel anything... Because it [the vaccine] is not something that I need to talk or say something about, because it is something that I don't believe, even the symptoms of it, I never had them, and I am not vaccinated but I am still alive.*
>
> *(Female FGD participant, Rural, South Africa).*

Other young people were still uncertain about the vaccine; in some cases this was linked to the belief that they did not have adequate knowledge about it.

> *I don't know everything about the vaccine so I cannot take a stand. I do not have adequate knowledge and am unsure of it.*
>
> *(Female IDI participant, Urban, South Africa).*

2. Understanding of the intervention

Two key subthemes emerged in relation to the *intervention understanding* component, defined as the extent to which young people have adequate knowledge and understanding of the intervention. These were: adequacy of the information and communication around the Covid-19 virus and vaccine; and conflicting information and misconceptions.

**Adequacy of information and communication.** There was disagreement, among adolescent and youth participants, as to whether information and communication around the vaccine had been sufficient and appropriate. Some young people lamented that they did not have sufficient knowledge of the virus and needed to be provided with more and better information. This was the more dominant perspective. This lack of knowledge was reported to be contributing to their negative opinion of the vaccine or their inability to form a clear opinion.

> *They must tell us about this disease, where it comes from because other diseases we know, like TB is spreading by coughing. But this one we don't know where it comes from and I don't understand if it is spreading by air or touching, if it exists or not.*
>
> *(Female FGD participant, Rural, South Africa).*

Young people suggested better use of media, such as television, internet and social media platforms, to better communicate more detailed and accurate information with young people.

However, there was some disagreement around who would be the most trusted individuals to relay key messages. For example, in one focus group conducted in rural South Africa, some young people indicated that they would be more likely to trust people from outside of their community, since outsiders would have more information; moreover, it would be hard to believe positive vaccine messages from the same community members spreading false information and denying the existence of the virus. Instead, other young people argued that they would prefer communications from trusted people in their community:

*...because if it is people from outside we won't believe them because we will not know if they are lying or they want us to go and get vaccinated.*

*(Female FGD participant, Rural, South Africa).*

A less dominant perspective, among a minority of young people, was that young people had sufficient knowledge about the Covid-19 vaccine, and that communication around vaccination had been good. Cited sources of information included clinic staff at vaccination sites, YouTube and other social media sources, and school staff.

*They [clinic staff] are welcoming, they explain everything to you as well as the different types of vaccines they have, and then you get to choose.*

*(Female FGD participant, Rural, South Africa).*

**Conflicting information and misconceptions.** Many young people highlighted their difficulty in the face of conflicting information from multiple sources, such as social media and community members. This resulted in confusion and fuelled distrust, since at times they weren't sure who to believe, or how to distinguish false from accurate information. This was also linked to misconceptions and misunderstandings about the virus and vaccine.

*In my opinion we are currently receiving different views about this vaccine, so we don't know who to listen to or to trust... I would love to vaccinate but I can't because of these different views. Some say 'we will die', others say they are installing coronavirus during the vaccination process.*

*(Female FGD participant, Urban, South Africa).*

3. Self-efficacy

The dominant idea that emerged in relation to self-efficacy, defined as young people's confidence in their ability to perform the behaviours needed to be vaccinated, was that vaccines were easily accessible. This indicated that the young people felt confident in their ability to do what was necessary to be vaccinated. Reasons included proximity and easy access to vaccine sites, being able to be vaccinated at school or at mobile clinics in the community, the existence of multiple sites in different locations, and short queues and friendly, helpful staff at clinics.

*I think it is convenient because many people don't go, so there is not a long queue. So it is easy if you just walk into the office, they will just give you the vaccines and I think they give a certificate also.*

*(Female IDI participant, Urban, Nigeria).*

4. Burden

The two themes emerging for 'burden', referring to the experienced or anticipated costs and efforts related to vaccination, were constraints related to reaching and accessing the vaccination sites (i.e., distance and transport), and constraints at the sites or facilities. The majority of participants raising these challenges were residing in Nigeria or rural areas of South Africa.

**Distance and transport to vaccination sites.** For those young people who lived far from health facilities or other vaccination sites, and/or had to take public transport, the costs and effort associated with accessing the clinics were considered obstacles to vaccination. This was exacerbated by having to return multiple times to the site, for multiple doses or because of being turned away the first time. Mobile clinics could help address lack of facilities to an extent, but were only available on certain days.

> *Like, I'm staying … where I'm staying from Achuza here area to Makurdi local government, I paid 300 Naira going there and coming back, and lets just assume that you went there the first day... they will not see them and you know, the next day, the cost of transport, coming the next day again. They will be like, they will get upset now.*
>
> *(Male IDI participant, Urban, Nigeria).*

**Constraints at vaccination sites.** A minority of young people complained about the conditions and waiting time at the facilities where vaccines were made available. These included long queues, poor service and unfriendly staff and, in some cases, being turned away unvaccinated and told to return another day.

> *No, it wasn't easy for me because the services there were very bad. We came there early around seven o'clock... There was nobody there, we just queued up, queued, so it was around nine and I was having classes that day also. I went late. Their services were very poor, yes. They were also rude which is not so good.*
>
> *(Male IDI participant, Urban, Nigeria).*

5. Relevance

Relevance refers to the extent that young people considered Covid-19 vaccination to be in line with their needs and lived experiences. The three themes that emerged in relation to relevance were: 1) the belief that the virus does not exist; 2) the belief in low individual vulnerability to the virus and; 3) whether or not Covid-19 vaccination was a priority in the young person's life.

**Belief that Covid-19 doesn't exist.** Some young people questioned the very existence of the virus. This spoke directly to (lack of) relevance since a vaccine for what is considered a non-existent disease could not be considered useful in a young person's life. This theme was raised exclusively among young people from rural South African communities participating in focus groups, and did not appear to reflect the position of the majority of young people.

> *So, I cannot get vaccinated whilst I strongly believe that Covid-19 is not real.*
>
> *(Female FGD participant, Rural, South Africa).*

**Belief in low vulnerability to Covid-19.** Other young people believed that the virus existed, but that they were not vulnerable to contracting it. Similarly, the belief in low personal vulnerability implied that vaccination would not be particularly useful or relevant to these young people's lives. This belief was expressed by a minority of mainly male participants.

*There is no need. Ever since the breakout, I have never been affected at all and I don't see how it will ever affect me in the near future.*

*(Male FGD participant, Urban, South Africa).*

**Vaccination as a priority in life.** There was disagreement among participants as to whether being vaccinated was a priority in their lives, with some young people arguing that it was, and others that it was not. Those who felt it was a priority referred to potential positive consequences of vaccination, such as access and opportunities that came with vaccination, as well as the need to protect oneself and one's family. A few participants felt this was especially important among young people, because they were a higher risk group that were more prone to socializing and moving around.

*I think it is important for us as youth, especially when we are gathered together in an indoor space having a party, that is treated as a super spreader. So getting vaccinated is so important since we can't be easily controlled.*

*(Female FGD participant, Rural, South Africa).*

Those who instead felt that Covid-19 vaccination was not a priority, cited various reasons for this. These included: preferring to spend time with friends, the fact that the prevalence rate was going down so there was no need (linked to perceptions of low vulnerability), no immediate plans to travel or access venues for which vaccination was a prerequisite, and a belief that the vaccine was not effective anyway.

*Its not a priority because I have like nothing to do, like its, you see like when it's a priority I have to maybe go out of the country then I have to do it before I go out of the country. So I think I am not going anywhere soon.*

*(Female IDI participant, Urban, Nigeria).*

A few young people had a fatalistic outlook, in that they held the belief that if they contracted the virus it was meant to be and unavoidable. Others had no clear, apparent reason but simply did not consider it important.

*I was not, I felt like I don't need a vaccine, I can just be myself. If anything happens to me that's fine, maybe that was what was supposed to happen...*

*(Female IDI participant, Urban, South Africa).*

6. Ethicality

Two key themes emerged from the data with regard to ethicality, defined as the alignment of the Covid-19 vaccination intervention with young people's value systems, and/or that of their families and communities. These were: alignment with religious beliefs and/or groups; and alignment with traditional beliefs.

**Alignment with religious beliefs and groups.** There were diverging perspectives as some participants believed that vaccination was aligned with the values of their faith group or beliefs, while others believed it conflicted with these. Some young people believed that only God could determine whether they would contract the disease and only God could save them from it.

*It does clash somehow. Most churches believe that they should not do blood transfusions and they go against vaccines. They believe that if they get infected with Corona they should pray and everything will be okay.*

*(Female FGD participant, Urban, South Africa).*

An alternative (though isolated) perspective offered was that these various beliefs or values could co-exist.

*There are two beliefs system here, it is about trusting in God but you also need to get vaccinated in order to prevent yourself from contracting the virus. I combine both situations into my life, because I can't just be a one-minded person.*

*(Female IDI participant, Urban, South Africa).*

**Alignment with traditional beliefs.** A few young people raised the issue of the conflict between their traditional beliefs and vaccination (as a 'Western' biomedical intervention). They suggested that traditional or natural remedies were the better approach to preventing Covid-19, while not explicitly referring to this conflict. However, this was not a dominant theme, as it was raised only by a few participants.

*I believe in in my ancestors, you understand? Yeah, in traditional medicines, yes. You know, in everything that is African and everything that you know, that is my roots. Sometimes taking, you know, Western things is kinda like, 'Oh, my God'.*

*(Female IDI participant, Urban, South Africa).*

7. Perceived negative effects

The two key themes that spoke to perceived negative effects of the Covid-19 vaccine were side effects of vaccination (experienced, observed or anticipated) and fear of needles.

**Side effects of the vaccine.** The most frequently raised concern with the Covid-19 vaccine was fear of side effects. In some cases these fears reflected 'real' or possible side effects, based on what young people had experienced or observed in others.

*With side-effects I experienced after getting vaccinated I almost lost my life and it was hard cause exams were about to kickstart... I would feel dizzy, vibration and sneezing like hell, and you can't just take anything like pain killer tablets for a headache to treat yourself, as the vaccination rules and regulations state it clearly. It was so hard to endure it I must say.*

*(Female FGD participant, Urban, South Africa).*

In other cases, these concerns were fuelled by myths and misinformation regarding possible effects of the vaccine. Many young people believed that the vaccine would cause certain death, for example.

*So, first of all, I am afraid of the pain. Secondly, on social media they say those vaccinated will only have 5 years to live from vaccination date to death...*

*(Female FGD participant, Rural, South Africa).*

There were also references to differential effects, i.e., the vaccine leading to bad side effects for some people, but not for others.

*... sometimes the vaccine does not make you sick if your immune system is strong, but to other people the vaccine has side effects and they get really sick.*

*(Female FGD participant, Urban, South Africa).*

**Fear of needles and related pain.** A further reason many young people gave for not wanting to be vaccinated was their fear of needles and the anticipated pain from the injection. It was suggested that if the vaccine could be replaced by a pill, people would be more likely to take it up.

*People are saying different things and those things make me afraid. Some people are saying it is right to vaccinate, while others say it is wrong. I also prefer a pill. I am scared of injection.*

*(Female FGD participant, Urban, South Africa).*

8. Perceived positive effects

The three emerging themes related to perceived positive effects of the vaccine were: 1) perceived effectiveness of the vaccine in reducing infection and death; 2) protection of others; and 3) resumption of 'normal' life.

**Perceived effectiveness in reducing infection and death.** A significant proportion of young people believed that the vaccine was effective in reducing the transmission of Covid-19 and number of infections, as well as reducing the likelihood of death once one had contracted the virus.

*What I am saying is, getting vaccinated is something which is good. Because ever since the vaccine was introduced, the infection rate started dropping bit by bit. If everyone can take a jab, perhaps it may eventually be defeated entirely.*

*(Female FGD participant, Rural, South Africa).*

However, participants were overall divided on this point, as perspectives differed with some young people arguing that the vaccine was ineffective. The main reason given was that they had observed people they knew, and high-profile people, contracting Covid-19 despite being vaccinated. For many, the fact that the vaccine was not 100% effective in preventing infection rendered it ineffective in their opinion.

*Even Ramaphosa [President of South Africa] vaccinated but later he was infected with the same virus.*

*(Female FGD participant, Urban, South Africa).*

A minority of young people were still uncertain as to whether the vaccine was effective or not; this appeared to be linked to inadequate understanding. There were also references to differential effectiveness with the vaccine being effective for some but not others. Participants suggested this could be a result of underlying conditions in some individuals, different immune systems or simply different effects of medication on different people.

*It is possible that I may have some underlying health condition, but I am not yet aware of it. Then when I go to get vaccinated, the vaccine might trigger something in me and then I become ill, whereas, with some who have good health, nothing happens.*

*(Female FGD participant, Rural, South Africa).*

**Protection of family members and others.** More broadly, the vaccine was seen by many young people as a means to protect their family or other people that would come in contact with them. There were several references specifically to protecting grandparents and other older family members, who were the most vulnerable.

*Because our grannies are much weaker than us, so it will be important for us to get vaccinated so that we won't affect them, because if they are infected they will get sick and die.*

*(Male FGD participant, Rural, South Africa).*

A few young people took a broader approach to this issue, believing that vaccination was a social responsibility, to contribute to herd immunity and eradicating the virus.

*I didn't have necessarily a gun pointed towards my head to get the vaccine from other people. I just felt it was maybe important to do, just to get the herd immunity going, contribute towards that.*

*(Male IDI participant, Urban, South Africa).*

**Resumption of 'normal' life.** A third theme, related to positive effects of the Covid-19 vaccine, referred to vaccination contributing to the ability to resume 'normal life' and being subjected to fewer restrictions. This included being able to access workplaces and learning institutions, being able to travel and to resume a social life. The possibility of not having to take other preventative measures, such as mask wearing, was also seen as a potential positive consequence of vaccination. Some young people indicated that the only thing that could coerce them into vaccination was access to places or opportunities, such as job opportunities or travel.

*The only thing that can make me vaccinate is none other than job opportunities... Or if possible I would buy false proof.*

*(Male FGD participant, Urban, South Africa).*

9. Perceived social acceptability

The last component of our model is social acceptability, defined as young people's perceived acceptability of the intervention by others. The three main themes that emerged were 1) the idea that the decision of whether or not to vaccinate belonged only to the young people, who should make it independently from social influence; 2) social opposition (or lack of social acceptability) to the vaccine and consequent negative influence; and 3) social acceptability and positive social influence.

**Independence from social influence: Decision belongs to young people.** Some young people suggested that they made decisions independently from social influence and/or that the decision of whether to be vaccinated should be theirs alone. Reasons given were that community members would criticize them regardless, and that it was the young person who would have to bear the consequences of their decisions.

*Community members are prone to criticise, that's why I believe in making my own decisions.*

*(Female FGD participant, Rural, South Africa).*

**Lack of social acceptance and negative social influence.** Many young people spoke of opposition to vaccination, and stigmatization of those who were vaccinated, among their

friends and family. In many cases this was linked to myths and misconceptions around the vaccine, for example the belief that vaccination could lead to death and increased risk of transmission. In some cases, participants indicated having been vaccinated or being willing to be vaccinated despite negative social influence. There were even a few references to young people having to hide the fact that they had been vaccinated from their parents, elders, peers or other individuals in their lives. In other cases, participants suggested that negative social influence discouraged them from vaccination.

*I think in my community there will be a lot of judgmental comments as they won't understand why I vaccinated, as they say that the vaccine kills people. Others will say, now that I have vaccinated, I am the one that is spreading the virus.*

*(Female FGD participant, Urban, South Africa).*

**Social acceptance and positive social influence.** Other young people indicated that their friends, caregivers, family and/or faith communities would accept their decision to be vaccinated, and in some cases even encourage or support it. A few young people even suggested that their parents and those of other young people would likely support them logistically and financially, to encourage them to be vaccinated.

*Yes. They're [my family members] vaccinated. Mum even forced me, I was like scared of, but if in fact, when she confronted me that I must go and do it, I had to lie to her that I collected it already. But later on she asked me to bring the vaccine, the card... that was where I was caught, I couldn't provide any means to prove that I was like vaccinated. So, I had to like to follow them.*

*(Male IDI participant, Urban, Nigeria).*

Conversely, a few young people felt that they could have a positive influence on their friends or family, through their behaviour being modelled.

*Yes, it can have a positive impact because it will not be easy for the virus to infect me [after being vaccinated], and they will also realise they should go and get vaccinated themselves.*

*(Male FGD participant, Rural, South Africa)*

## Discussion

This paper makes a key contribution to the existing literature on vaccine acceptability, and acceptability more broadly, by applying a recently published conceptual framework to better understand this important global public health issue among a high-risk population group for vaccine hesitancy (young people) in two African countries. Our findings provide insight into the reasons as to why young people are willing or unwilling to be vaccinated, and the sources influencing their decision-making [57]. They show that vaccine acceptability or hesitancy is shaped by a multiplicity of complex and inter-related factors, as highlighted by other research studies from Africa and beyond [8,33]. Moreover, they highlight the relative importance of these factors, as well as providing a more in-depth perspective of some of these phenomena and their connections.

### Challenges to vaccine acceptability

Some young people discussed challenges related to transport to and constraints at vaccine sites. These challenges were raised mainly by participants residing in Nigeria and rural South

Africa and should be considered in light of the more limited coverage of the Nigerian vaccination campaign at the time of data collection (early 2022). Nevertheless, it appears that overall the 'burden' (costs, efforts and opportunity costs) associated with accessing vaccines was not a key obstacle to vaccine acceptability and uptake for these young people. This is noteworthy considering that this research was conducted with individuals from resource-deprived communities and mainly resource-deprived households. Similarly, issues related to self-efficacy, including confidence in being able to access vaccine sites and go through with vaccination, did not emerge as dominant factors explaining low acceptability. Instead our study exposes limited vaccine understanding, conflicting information and distrust, the influence of others, as well as fear of side effects, as the key inter-related factors driving low vaccine acceptability and willingness to be vaccinated among these young people [58].

Many of the factors influencing low Covid-19 vaccine acceptability identified in this sample have been found by quantitative studies, conducted with young people in Africa and other parts of the world; these include inadequate information, conflicting media information, religious beliefs, low perceived infection risk, concerns about vaccine safety and effectiveness, and low trust in public health institutions [30]. These factors also resonate quite closely with factors associated with Covid-19 vaccine hesitancy found by recent studies with young people in other (mainly high income) countries, which have also shown previous vaccine history (e.g., for the influenza vaccine) to be a key determinant [59–64], and with factors driving low acceptability of other vaccines, such as the HPV vaccine, among African adolescents and youth [40]. These common and recurring themes indicate that it would be important to address certain attitudes and beliefs to increase vaccine acceptability more broadly - not only to be better prepared for future infectious disease outbreaks, but also to ensure uptake of routine vaccinations.

As indicated above, our findings also provide a more in-depth perspective of some of these phenomena and their interaction in this population group. For example, knowledge, (dis) trust and information sources appeared to be central to acceptability of Covid-19 vaccination, and to be interlinked. The lack of understanding of the disease and the vaccine in this group of young people, across countries and (urban/rural) geographical locations, was noteworthy, particularly when considering public health messaging policies and media campaigns conducted in each of these countries through channels such as radio and television, posters and digital media [65–67]. While technology and social media platforms can be used as a tool for information dissemination, they can also amplify misinformation and information overload during an infodemic [68,69], and clearly in some cases the latter may dominate.

Our findings suggest that we need to find more effective ways of engaging with young people to allay myths and misinformation and help them navigate the barrage of conflicting information they are being exposed to [8,70]. This will entail messaging that they can understand and relate to their lives and needs, based on what is of most value to them [68]. It may involve potential two-way communication with opportunities to engage versus simply providing further information. Moreover, resources should be leveraged to promote vaccination more broadly, not only for Covid-19 but also routine vaccinations. These may include cross-sectoral initiatives that employ community outreach, multimedia content and social mobilization, mobile messengers and pop-up vaccination sites, and youth-led interventions such as community radio programmes [71,72]. Health promotion and education specialists, as well as young people themselves, should be central to these efforts; these cannot be left primarily to biomedical experts.

Our findings also suggest that accurate information in itself may not be enough. For outreach to be effective it is important to understand not only which channels young people are using to obtain information but also *who* is trusted and best suited to provide it in a given

context. This is all the more relevant given the central role of perceived social acceptability for some young people's decisions. For example, a survey conducted remotely with adolescents in 5 sub-Saharan African countries found vaccine acceptability to be most influenced by healthcare workers, parents or family members, and schoolteachers [30]. Some authors have proposed strategies operating simultaneously through multiple platforms, including government-NGO collaborations, popular social media platforms, the use of traditional media (e.g., radio, television, magazines), engagement of opinion leaders in society, and the use of existing channels such as community groups [68].

As our data shows, trusted sources may also differ across contexts and groups of young people, which is why it is important to seek input and advice from young people themselves. It is also noteworthy that most of the young people involved in this study were participating in programs run by trusted organisations (some of which had been working in the community for years, even decades). Adding components to existing community organisation programs or networks could provide an opportunity to channel accurate information on Covid-19 vaccination from trusted sources. This is something for funders and programme planners to further consider in the future.

Moreover, young people's distrustful attitudes towards vaccination have no doubt also been influenced by a wider context of waning trust in governments and disengagement of young people from political life [8,48,73,74]. While changes in messaging and policy may be essential and advisable, in a situation where a virus and our knowledge about it is changing rapidly, this may be viewed – as our findings show – as inconsistency, incompetence and even deceit; no doubt worsened by allegations of corruption related to vaccine roll-out in certain contexts [8,75]. Trust in health systems and policymakers has been shown to be key to ensuring adherence to public health measures [76], as has a sense of national identity [77]. Governments and other stakeholders need to find better ways to regain trust among young people, and to better manage and communicate the inevitable variability of policies and information during public health crises. Cooper et al argue, for example, that more timely, transparent communication on vaccine-related decisions is needed to address concerns as they come up, taking into account local beliefs and norms [17]. More broadly this entails addressing governance issues beyond health, that operate at a global and regional level and influence health programming [78].

## Factors encouraging vaccine acceptability and uptake

Our findings also point to factors that may encourage or enable Covid-19 vaccine acceptability. Some of these are highlighted by recent studies with adolescents and youth in other parts of the world (mainly high-income countries). These motivators include a desire for protection from disease, positive perceptions of vaccine safety and efficacy, a desire to return to normal activity, a desire to protect known others and, in some cases, even a sense of broader social responsibility [59–64]. Once again these findings resonate with reasons given for acceptability of other (HIV or HPV) vaccines among young people in Africa [40]. This reinforces the possibility of entry points for intervention, again not limited to Covid-19 vaccination, but potentially to better leverage available resources for immunization and vaccination promotion more broadly.

Initiatives could include family- and peer- oriented programs and campaigns that appeal to a sense of social responsibility and willingness to engage in collectively-oriented actions [77]. Taking a family or community approach to promoting vaccine acceptability aligns with the communal values and collective decision-making embedded in many African societies [79]. Young people's communication with caregivers and family has also been found to be a predictor of intention to vaccinate [80] and trust in government [73]. Similarly, campaigns should recognize

the influence of peers among young people, as they often play a significant role in shaping attitudes and behaviours, and involve individuals who could be role models [81,82].

Additionally, young people's agency in believing they can be a positive influence on others, particularly in the realm of vaccine acceptability, plays a crucial role in fostering a culture of public health awareness. The enthusiasm and energy of young people can be a powerful force in shaping perceptions and attitudes towards vaccinations. As social models able to connect with peers through various platforms, young individuals can engage in open conversations, share accurate information, and debunk myths surrounding vaccines [83,84].

## Areas of uncertainty and disagreement among young people

Our findings also provide further depth on the nuances and complexities of some of the phenomena shaping Covid-19 vaccination among these young people, by highlighting differing perspectives. Although negative attitudes towards the vaccine were dominant, our findings also highlighted positive and indifferent attitudes towards the vaccine among other young people, differing beliefs around sufficient knowledge and vaccine effectiveness, and differing perspectives on social acceptability linked to vaccination. These differences remind us that young people are not a homogeneous group and suggest that different approaches may be needed to address different groups of young people with different reasons for hesitancy. They may also represent an opportunity for intervention, specifically with young people who are undecided or uncertain [17,21].

There were also certain 'tensions' or apparent contradictions in the data. For example, on the one hand many young people indicated that mandatory vaccination for access to learning institutions, employment, travel and other services or venues was the only reason they had been vaccinated for Covid-19 or would consider vaccination. Yet at the same time participants expressed unhappiness about "being forced to be vaccinated" in order to access certain venues and services, and believed this had violated their human rights and compromised their trust in public institutions. During the Covid-19 epidemic, countries and key development organisations have grappled with the dilemma of whether or not to enforce mandatory Covid-19 vaccination, where the need to guard human rights and avoid heightening government distrust had to be counterposed with the need to protect populations from infection [8,85,86]. The urgency of decision-making appears to have passed for Covid-19 variants, and clearly the need to avoid infringing human rights has prevailed overall. However, scientists, policymakers and other key stakeholders need to be prepared to engage once again with these difficult questions when they rear their head during future outbreaks of infectious disease.

Also worth noting was the 'all or nothing' binary attitudes towards vaccine effectiveness among our participants, where the vaccine was either considered completely effective or simply 'ineffective' because effectiveness was not 100%. From this perspective the case of even one known vaccinated person contracting the disease (such as the South African president) was taken to be evidence of the vaccine being ineffective and not worthwhile. The concept of some level of partial effectiveness (reduction of risk of infection, symptom severity or mortality) being better than nothing did not seem to prevail in this sample. This finding points to the need to find a way of conveying more nuanced messages around effectiveness so that young people can better relate to the concept of risk reduction being a worthwhile objective of vaccination more broadly. Once again, this highlights how communicating key health promotion messages can be difficult, and that we need to involve different professionals, including health promotion specialists, and young people themselves. We have highlighted this point, and other key recommendations for policymakers and practitioners emerging from our findings, in Table 1.

**Table 1. Recommendations for policy and practice.**

| 10 recommendations for policy and practice |
| --- |
| 1. **Address common attitudes and beliefs fuelling vaccine hesitancy, such as:**<br>• inadequate and conflicting information<br>• low perceived infection risk<br>• concerns about vaccine safety and effectiveness<br>• low trust in public health institutions |
| 2. **Involve young people and health promotion specialists in health communication and promotion** |
| 3. **Leverage resources to promote vaccination not only for Covid-19 but also routine vaccinations** |
| 4. **Determine *who* is trusted by young people and best suited to provide information in a given context.** |
| 5. **Work with multiple simultaneous strategies and platforms such as:**<br>• government-NGO collaborations<br>• popular social media<br>• traditional media<br>• opinion leaders and community groups<br>• social mobilization mobile messengers<br>• pop-up vaccination sites<br>• youth-led initiatives such as community radio |
| 6. **Link health promotion messaging to trusted local organisations and programs already working in communities** |
| 7. **Find ways to increase young people's trust in institutions, through more timely and transparent communication on health crises and initiatives** |
| 8. **Use enablers of Covid-19 vaccine acceptability as entry points for intervention:**<br>• desire for protection from disease<br>• positive perceptions of vaccine safety and efficacy<br>• a desire to return to normal activity<br>• a desire to protect others |
| 9. **Consider family- and peer- oriented approaches to programs and campaigns** |
| 10. **Recognise young people's agency in fostering awareness and behavioural change linked to vaccination** |

## Reflections on the accelerate model

Given that this is our first application of the *Accelerate* model to empirical data, it is worth briefly reflecting on our experience of using this framework. Our framework seemed to work well to elucidate and structure the various factors shaping acceptability. It also appeared to be comprehensive as all emerging themes in the data fit well within at least one of the model components, as subthemes; there was no need to add further components or base themes.

It was clear that there were linkages between the various components of the model. For example, sufficient or insufficient information or knowledge about the vaccine, beliefs in effectiveness, and concern with potential negative effects (e.g., side effects) were among factors influencing (positive or negative) affective attitudes. Social acceptability was linked to myths and misinformation. These linkages support our hypothesis of acceptability as an emergent property of a complex, adaptive system of interacting components, which can both influence and be influenced by user engagement [6,87]. The interaction between components is important to bear in mind both for conceptual work (e.g., when developing this framework into a behavioural model) and for empirical work (e.g., to explore potential causal relationships and linkages between model components in specific contexts and populations) [6].

Lastly, there was some overlap in our findings across themes and subthemes, in part due to links between factors mentioned above. In particular, *affective attitudes* appeared to be shaped by many of the other components in the model. This suggests that, should researchers need to use a shorter tool, intervention 'rating' and an open question focusing on attitudes towards an

intervention and reasons explaining these, would still be valuable and likely capture most of the key factors influencing acceptability.

## Study limitations

This study also has a number of limitations. As indicated above, this was an exploratory study with a convenience sample, conducted within a limited timeframe and with limited resources. The study design and sub-sample sizes did not allow us to properly explore differences in perceptions across race, gender or location (urban versus rural or across countries). Initially, we intended to differentiate participants by age, to identify similarities and differences in perspectives between older adolescents (15-19) and youth (20-24). However, we encountered challenges linked to recruitment conducted through the CBO partners, and logistical constraints, which did not allow us to achieve this disaggregation. We recognize this as a limitation and an opportunity for future research. More specifically, we note that the majority of adolescents in our sample were from one province in South Africa (KwaZulu-Natal) and that their perspectives may therefore be disproportionately reflected in this data. This was, once again, a result of a sampling process guided primarily by access and convenience, based in part on the locations of the researchers and CBO partners, and availability of community-based organizations (CBOs) to partner with us and assist in the recruitment process within a given timeframe.

Since the study relies on self-reported data, some collected in a group setting, there is also the potential for social desirability bias. As indicated above, we should also acknowledge and consider the potential role of parents and guardians in influencing young people's decisions and behaviour in relation to vaccination uptake, especially among younger adolescents. We also acknowledge that cultural differences and differences in country policies, communication strategies, and vaccine rollout across countries could have affected participants' perspectives; the study was not able to determine the extent of these influences. Lastly, despite the measures taken to ensure reflexivity (described above), the researchers' social, cultural, and personal positions may have influenced how they approached the study, selected participants, and interpreted the findings.

## Conclusion

Low vaccine acceptability in the context of Covid -19 can be seen as an acute – but not isolated – phenomenon within an alarming longer-term trend of broader vaccine distrust in Africa. This trend started well before Covid-19 and will not end with this virus, although there is some evidence that it may have been heightened by weaknesses in the Covid-19 global response. Low vaccine acceptability poses a serious risk for public health in general, as well as the success of future public health initiatives aimed at controlling infectious disease outbreaks. As other authors have argued, attention should be paid to particularly vaccine-hesitant and high-risk populations, such as adolescents and youth [30]. Campaigns and interventions should be targeted to their concerns and needs.

Covid-19 has left the world with considerable primary and secondary negative consequences, at a time when trust in government, health authorities and systems in Africa is already compromised. An effective response to this situation may require a shift in our thinking to see this health crisis as a unique opportunity to promote vaccine acceptance, transparency and public trust more broadly [17,88]. Now that the acute crisis phase of Covid-19 has passed, there is the understandable risk of neglecting vaccine acceptability as we shift attention to current priorities. This would be a mistake. We know that the moments of acute crisis, such as the early stages of a rapidly spreading new virus, are not effective times to address distrust of the very health tools needed to urgently control disease. Instead, there is a need for ongoing and longer-term efforts

                                        

to encourage vaccination and build trust more broadly among higher risk populations such as adolescents and youth. These efforts should focus on the factors, highlighted above, that appear to shape vaccine acceptability among young people across different contexts and diseases. They should also include the development, application and iterative refinement of frameworks and tools, such as the *Accelerate* framework, to better understand and assess acceptability and its components. Now is the time to be acting, before the next global infectious disease crisis is already upon us. Governments, international development agencies, scientists, NGOs and community organisations all have a role to play to make sure this happens.

## Supporting information

**S1 Table. Participant characteristics.**
(DOCX)

## Acknowledgments

We acknowledge the important contribution of our community-based partners in South Africa and Nigeria as well as the field researchers who worked on data collection in KwaZulu-Natal, South Africa. We are very thankful to all the young people who shared their time and perspectives with us.

## Author contributions

**Conceptualization:** Marisa Casale, Oluwaseyi Somefun, Genevieve Haupt Ronnie, Olagoke Akintola, Lorraine Sherr, Lucie Cluver.

**Data curation:** Marisa Casale, Oluwaseyi Somefun, Genevieve Haupt Ronnie, Joshua Sumankuuro.

**Formal analysis:** Marisa Casale, Oluwaseyi Somefun, Genevieve Haupt Ronnie, Joshua Sumankuuro.

**Funding acquisition:** Lorraine Sherr, Lucie Cluver.

**Investigation:** Marisa Casale, Oluwaseyi Somefun, Genevieve Haupt Ronnie.

**Methodology:** Marisa Casale, Oluwaseyi Somefun, Genevieve Haupt Ronnie, Joshua Sumankuuro, Olagoke Akintola, Lorraine Sherr, Lucie Cluver.

**Project administration:** Marisa Casale, Oluwaseyi Somefun, Genevieve Haupt Ronnie.

**Supervision:** Marisa Casale, Oluwaseyi Somefun, Olagoke Akintola.

**Validation:** Marisa Casale, Oluwaseyi Somefun, Joshua Sumankuuro.

**Writing – original draft:** Marisa Casale, Oluwaseyi Somefun.

**Writing – review & editing:** Marisa Casale, Oluwaseyi Somefun, Genevieve Haupt Ronnie, Joshua Sumankuuro, Olagoke Akintola, Lorraine Sherr, Lucie Cluver.

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
