## [Decision Letter · Decision Letter 0]

11 Jul 2024

PGPH-D-24-01282

Factors shaping Covid-19 vaccine acceptability among young people in Nigeria, South Africa and Zambia: An exploratory qualitative study

Dear Dr. Casale,

Thank you for submitting your manuscript to PLOS Global Public Health. After careful consideration, we feel that it has merit but does not fully meet PLOS Global Public Health’s publication criteria as it currently stands. Therefore, we invite you to submit a revised version of the manuscript that addresses the points raised during the review process.

We look forward to receiving your revised manuscript.

Kind regards,

Bey-Marrie Schmidt, PhD

Academic Editor

Journal Requirements:

3. In the online submission form, you indicated that "We are happy to make excerpts of the qualitative data transcripts relevant to the study available upon request. The theoretical model used to guide the study has been previously published in an open access publication.". 

3. Uploaded as supplementary information.

Reviewers' comments:

Reviewer's Responses to Questions

**Comments to the Author**

1. Does this manuscript meet PLOS Global Public Health’s publication criteria ? Is the manuscript technically sound, and do the data support the conclusions? The manuscript must describe methodologically and ethically rigorous research with conclusions that are appropriately drawn based on the data presented.

Reviewer #1: Yes

Reviewer #2: Partly

2. Has the statistical analysis been performed appropriately and rigorously?

Reviewer #1: N/A

Reviewer #2: N/A

3. Have the authors made all data underlying the findings in their manuscript fully available (please refer to the Data Availability Statement at the start of the manuscript PDF file)?

Reviewer #1: Yes

Reviewer #2: Yes

4. Is the manuscript presented in an intelligible fashion and written in standard English?

Reviewer #1: Yes

Reviewer #2: Yes

5. Review Comments to the Author

Reviewer #1: This paper addresses an important topic and the findings are consistent with other studies addressing vaccine hesitancy/acceptance and therefore the findings contribute to a growing body of evidence. The focus on young people in Africa is particularly valuable. The authors use their own "Accelerate Framework for Young People's Acceptability" to explore their findings and this is a novel contribution in a field that is dominated by well established models for vaccine acceptance. However, the paper is overly constrained by the application of the framework and does not pay sufficient attention to the context in which the research was being conducted. This led to a weakness in both the approach to analysis and in the discussion of the findings.

The study used a qualitative methodology including focus group discussions in rural & urban South Africa and key informant interviews with young people across three countries, South Africa, Zambia and Nigeria. Although research participants were engaged with the same research questions, overall the thematic analysis across the three countries felt awkward because of the scant attention to context. For example, although there is a date (early 2022) when the research was conducted there isn't a description of the COVID-19 vaccination services in any of the countries and when and how young people were invited to be vaccinated and how this may have shaped their responses. In South Africa there are comprehensive vaccination records that cover all regions which show the penetration of the vaccination campaign. The paper didn't discuss the vaccination status of the young people at the time of the data collection and I wondered if there was a reason for this.

Models around vaccine hesitancy/acceptance are important, however I did not feel convinced that the factors concerning vaccine acceptability should be framed outside of context. Given the large number of young people (n=127) participating in the South African focus groups it felt the researchers had an opportunity to explore the application of the framework to young people (from both urban and rural settings) within the context of South Africa. Many quotations are taken from this group of respondents too. Overall the research participants are largely from South Africa (n=136). Only two research participants are from Zambia. There are no quotes from Zambian respondents included in the analysis. The research feels uneven and it is suggested that Zambia be omitted from a reworked paper. The inclusion of data from respondents who participated in in-depth interviews in Nigeria and South Africa should also be considered in the light of comments above. For example, are these young people mainly from urban environments? And what was the context for youth vaccination in Nigeria at the time of data collection? It may be a cleaner data set if the paper focused on South Africa only.

In the introduction to the paper the authors make mention of the specific complexities related to vaccine acceptance by adolescents. However the analysis presented in the findings does not differentiate between respondents aged 15-19 and those aged 20-24 which also felt like a missed opportunity. In the descriptor for quotations there is no differentiation made between focus group participants coming from rural or urban contexts either, although there were more urban:rural participants than male:female participants which is used as a descriptor.

Reviewer #2: I enjoyed reading through this application of the recently designed theoretical framework to thrash out and understand Covid-19 vaccine acceptability in young people in three African countries. It agree with the authors that this provides contextual insights into acceptability as a construct, as well as specifically vaccine acceptability more generally. I’ve made a few comments or recommended changes below, with primary concerns related to evidence presented in the manuscript on methods:

Introduction:

- Clear problem formulation. I would steer away from using the term ‘assess’ in the aim, as the framework nor the qualitative methods appeared to be used to assess perspectives/are assessment tools. Rather it appears that this paper is looking at exploring or understanding this phenomena of acceptability, based in the explorative qualitative design.

- As a non-specialist with regards to vaccines hesitancy and acceptability, I found the use of terms (i.e. hesitancy, uptake, acceptability, willingness to be vaccinated etc) unfamiliar initially when reading. The second paragraph in the introduction was very useful to operationalise the terms being used within the manuscripts, and this could be expanded on for non-specialists.

- Further information on the “broader qualitative study” feels missing and how this particular sub-study fits within this. Specifically, to provide context and a better understanding of why methods were chosen. Include in either methods or here.

Methods:

- Clear outlining of recruitment methods, administration of FGDs and IDIs (i.e. training, language, length of time, where it took place, topic guides, compensation etc), and data analysis, as well as the different researchers roles in these. The score card technique for data collection was an interesting one and I appreciated the later reflection in the discussion on its suitability for use within exploring acceptability. However, limited justification of methods leaves gaps in my understanding of why these methods were used.

- Study setting: Limited contextual information provided other than country names, and CBO’s being in rural and urban KZN. Thick descriptions of contextual information are highly necessary to assist with rigour and, specifically, transferability.

- Data collection: Why were both FGDs and IDIs used as data collection tools? How did you decide what method to use in each context, (i.e. why were FGDs were only conducted in RSA?). Why were the IDI’s in English, when the FGDs were in local languages, and how do you feel this impacted on the results? I would cut down on recruitment ifromation and provide this information in more detail.

- Sampling: Although the recruitment methods were clearly outlined, sampling methods were not as evident, which was also recognised in the limitations. Most participants were sampled from rural and urban areas of KZN [n=136], but the reason for this wasn’t clearly outlined in the methods section nor the impact that this may have on the results.

o With so few participants from Zambia [n=2], I wonder whether including their results in the analysis without separating out their unique views somewhere in the analysis is the most rigourous approach, given the larger representation of South African and Nigerian perspectives. What strategies were put in place to ensure that perspectives from both Nigeria, and particularly Zambia, weren’t ‘drowned out’? Would it not be more rigourous to present this paper as findings from Nigeria and South Africa?

o In your limitations you mentioned the use of convenience sampling not being ideal, which I agree with, but justification for use of convenience sampling wasn’t clear?

- Analysis: Analysis seemed to follow methods related to Template Analysis or more general codebook thematic analysis, as I don’t see the specific steps related to charting, indexing and mapping that are unique and specific to FA. I would include evidence of these steps unique to FA or otherwise feel free to review Braun and Clarke’s helpful guide regarding different types of thematic analysis to see if King’s Template analysis (https://research.hud.ac.uk/research-subjects/human-health/template-analysis/) or codebook thematic analysis is a better fit in terms of your process followed (Braun, V., & Clarke, V. (2023). Toward good practice in thematic analysis: Avoiding common problems and be (com) ing a knowing researcher. International journal of transgender health, 24(1), 1-6.)

- Rigour: Rigour in data collection and analysis seemed clear through the use of co-coding, peer debriefing etc. Although the influence of researcher subjectivity was described in the limitations, I found limited evidence provided for how researcher reflexivity that was used as a rigour strategy to bring these biases to the forefront (for both researchers and the readers)? I would recommend including the strategies related to reflexivity in the manuscript.

Results:

- Personally I found that the use of subthemes with many quotes broke up the argument flow into too many ‘bits’ for the reader, but the subtheme headings provide an easy way to show the inter-relatedness of the various constructs and your arguments are well evidenced through the use of excerpts. The interrelatedness of different subthemes is clear and addressing in the reasons for this in the discussion was done.

Discussion:

- Good discussion points, particularly on the use of the framework, primarily based on evidence from results – I appreciated the investigation into the tensions within the dataset. Arguments within the discussion would ‘hit home’ better with further linkage to literature in the introduction, as some of the recommendations (although based on your evidence) are being introduced here for the first time here. I.e. There is a brief mention of interventions that have been used previously in the introduction, but building this evidence further in the introduction would make the arguments, for example, to include young people in development of interventions (which is mentioned three times in the discussion), stronger.

- I would also consider putting recommendations for interventions into a table/textbox for implementors as (i.e. important considerations for campaigns and vaccine messaging) are addressed in different aspects of the discussion – having one place where all this is placed for easy access could be useful.

- The paragraph about the partner CBOs not being involved in vaccine messaging (although valid) feels misplaced in the discussion. Part of this information would fit better in methods/ study setting and then built on here, included in the previous paragraph’s argument which also addressed using trusted individuals in dissemination.

- I found some language used (‘alarming’, ‘surprizing’) in paragraph 3 to describe the lack of intervention coherence somewhat borderline. While I appreciate the valid feelings of the researchers to this, this could be perceived as judgemental responses to challenges around intervention coherence, which is most likely due to inefficient communication strategies and misinformation, as was later stated. Similarly in the paragraph describing ‘all or nothing’ attitude towards effectiveness, there is presumption around the possible lack of understanding of the concept of partial risk reduction. Rather than implying this, I would report the findings and provide recommendations without presuming the underlying reason for this perception, as it wasn’t investigated.

- Lastly, regarding limitations, qualitative methods do not aim to be generalisable, but rather to provide rich and contextually relevant evidence around a phenomena, which you largely did provide - therefore there is no need to include this in the limitations.

Thank you for your wonderful work and all the best.

6. PLOS authors have the option to publish the peer review history of their article (what does this mean? ). If published, this will include your full peer review and any attached files.

**Do you want your identity to be public for this peer review?** For information about this choice, including consent withdrawal, please see our Privacy Policy .

Reviewer #1: No

Reviewer #2: No

---

## [Decision Letter · Decision Letter 1]

22 Nov 2024

PGPH-D-24-01282R1

Factors shaping Covid-19 vaccine acceptability among young people in South Africa and Nigeria: An exploratory qualitative study

Dear Dr. Casale,

Thank you for submitting your manuscript to PLOS Global Public Health. After careful consideration, we feel that it has merit but does not fully meet PLOS Global Public Health’s publication criteria as it currently stands. Therefore, we invite you to submit a revised version of the manuscript that addresses the points raised during the review process.

While the reviewers indicated that you addressed most of the comments that they provided in the last rounds of review, they have provided further queries that you will need to address. Consider addressing these comments that they have shared.

We look forward to receiving your revised manuscript.

Kind regards,

Ferdinand C Mukumbang, PhD

Academic Editor

Journal Requirements:

Additional Editor Comments (if provided):

Reviewers' comments:

Reviewer's Responses to Questions

**Comments to the Author**

1. If the authors have adequately addressed your comments raised in a previous round of review and you feel that this manuscript is now acceptable for publication, you may indicate that here to bypass the “Comments to the Author” section, enter your conflict of interest statement in the “Confidential to Editor” section, and submit your "Accept" recommendation.

Reviewer #1: All comments have been addressed

Reviewer #2: All comments have been addressed

2. Does this manuscript meet PLOS Global Public Health’s publication criteria ? Is the manuscript technically sound, and do the data support the conclusions? The manuscript must describe methodologically and ethically rigorous research with conclusions that are appropriately drawn based on the data presented.

Reviewer #1: Yes

Reviewer #2: Yes

3. Has the statistical analysis been performed appropriately and rigorously?

Reviewer #1: N/A

Reviewer #2: N/A

4. Have the authors made all data underlying the findings in their manuscript fully available (please refer to the Data Availability Statement at the start of the manuscript PDF file)?

Reviewer #1: Yes

Reviewer #2: Yes

5. Is the manuscript presented in an intelligible fashion and written in standard English?

Reviewer #1: Yes

Reviewer #2: Yes

6. Review Comments to the Author

Reviewer #1: The authors have carefully considered and responded to the comments of the reviewers and the paper is reading well. A few very minor points.

A final check for the use of adolescents, youth and young people to ensure consistency. A sentence describing the choice of terms in the methods would be appropriate.

The limitations of the study are very comprehensive and are very helpful. However a line in the methods under data analysis that describes why there is no country comparison or alternatively a reference to the section on limitations will facilitate the reader understanding this earlier in the paper.

Line 423 has 'the' missing in front of the word field.

The IDI respondents are better educated and have more opportunities. This distinction could be made clearer around line 450.

At line 655 under the theme burden I think it would be helpful to refer back to the period in which the data was collected and the limited coverage of the Nigerian vaccination campaign to provide context to the comments and perhaps a contrast with the South African experience.

Reviewer #2: Most of my comments have been addressed and the changes have made for a great article. Unfortunately, the presentation of the methods requires further minor work for consistency and clarity. Please see my attached comments.

7. PLOS authors have the option to publish the peer review history of their article (what does this mean? ). If published, this will include your full peer review and any attached files.

**Do you want your identity to be public for this peer review?** For information about this choice, including consent withdrawal, please see our Privacy Policy .

Reviewer #1: No

Reviewer #2: No

---

## [Editor Report · Decision Letter 2]

17 Dec 2024

Factors shaping Covid-19 vaccine acceptability among young people in South Africa and Nigeria: An exploratory qualitative study

PGPH-D-24-01282R2

Dear Dr Casale,

We are pleased to inform you that your manuscript 'Factors shaping Covid-19 vaccine acceptability among young people in South Africa and Nigeria: An exploratory qualitative study' has been provisionally accepted for publication in PLOS Global Public Health.

Best regards,

Ferdinand C Mukumbang, PhD

Academic Editor